# Characterization and Stability of a Novel Toxin in Scallop Mantle Tissue

**DOI:** 10.3390/foods12173224

**Published:** 2023-08-27

**Authors:** Nabuki Maeda, Takahiro Yumoto, Geng Xiong, Yasushi Hasegawa

**Affiliations:** College of Environmental Technology, Muroran Institute of Technology, 27-1 Mizumoto, Muroran 050-8585, Japan; nab.crystal.apr25@outlook.jp (N.M.); 10999537@mmm.muroran-it.ac.jp (T.Y.); 18748135846@163.com (G.X.)

**Keywords:** novel toxin, scallop, small intestine, subacute toxicity

## Abstract

Previous studies have shown that mice fed a diet containing 1% mantle tissue exhibited decreased food consumption and led to death. Toxic substances present in the mantle tissue have been isolated and identified. In the present study, we explored the characteristics and stability of mantle tissue toxicity. The treatment of mantle tissue with 1 mM hydrochloric acid, 1 mM sodium hydroxide, 1 mM dithiothreitol, and 1 mM hydrogen peroxide followed by heating did not significantly reduce the toxicity of mantle tissue in mice. These results suggest that mantle toxins are stable in tissues, particularly when exposed to acidic conditions and digestive enzymes. We examined whether mantle tissue exhibited acute toxicity. Mice fed a diet containing 20% mantle tissue did not show a distinct increase in toxicity compared with mice fed a diet containing 1% mantle tissue, demonstrating that feeding mantle tissue does not lead to acute toxicity. Finally, mantle tissue toxicity in the small intestine was examined. Chronic feeding of mantle tissue to mice changed the color of the small intestine. Real-time polymerase chain reaction analysis revealed that mantle tissue feeding caused changes in inflammation and endoplasmic reticulum stress markers in the small intestine. These results suggest that mantle tissue feeding causes toxicity after initial damage to the small intestinal tissue.

## 1. Introduction

Bivalve mollusks, such as scallops, mussels, and clams, accumulate toxins in their digestive glands by consuming toxic microalgae [1,2]. Paralytic, diarrhetic, and amnesic shellfish poisoning have been widely studied [3,4,5]. The most well-known causes of paralytic, diarrheal, and amnesic shellfish poisoning are saxitoxin, okadaic acid, and domoic acid, respectively. These toxins exhibit acute toxicity; however, there are no detailed reports on their subacute or chronic toxicity.

In Japan, scallop toxins are likely to occur during summer and fall, and paralytic and diarrhetic shellfish toxins are detected in different regions annually. In Japan, to minimize the risk of shellfish poisoning, the Ministry of Agriculture, Forestry, and Fisheries, as well as local authorities, conduct seawater monitoring and shellfish inspections. If shellfish toxins are detected, fishing and sales may be restricted, ensuring the protection of public health and enabling the safe consumption of shellfish.

Scallop is a significant marine product in Hokkaido, Japan, resulting in the generation of approximately 300,000 tons per year of scallop shells as industrial waste. For the effective utilization of the scallop shells, bioactivities of organic components in scallop shells have been extensively investigated [6,7,8,9,10]. Considering that these organic components are secreted from the mantle tissue, the bioactivities of the mantle tissue have also been investigated. In the process of this study, subacute toxicity was observed in mice fed a diet containing mantle tissue available on the market. Our previous study revealed that feeding mice a diet containing the mantle tissue of scallops led to increased levels of the liver damage markers aspartate aminotransferase and alanine aminotransferase, an increase in the kidney damage marker urea nitrogen, and subsequent death of the mice within a few weeks [11,12,13].

Furthermore, a novel scallop toxin comprising a complex of the N-terminal region of a gelsolin-like protein with a molecular weight of 18 kDa and actin fragments with a molecular weight of 25 kDa was isolated. Gelsolin-like protein is expressed in the mantle tissue, ovary, and gill tissue, but not in the adductor muscle, testis, or digestive glands [14]. Toxicity was only observed in scallop tissues expressing an 18 kDa gelsolin-like protein [14].

Although numerous proteinaceous toxins have been identified [15,16,17], little is known about those that exhibit toxicity after oral administration, because proteinaceous toxins are degraded by digestive enzymes in the gastrointestinal tract. In contrast, abrin and ricin in the seeds of the castor bean plant, *Ricinus communis* L (CB), which are isolated from plants, exhibit acute toxicity [18] even after oral administration. The subacute or chronic toxicity of these compounds has not been reported. However, the novel toxin isolated from scallops is proteinaceous and exhibits subacute toxicity even after oral administration [11].

Various types of shellfish, such as scallops, oysters, clams, and mussels, are widely consumed across Asia, Europe, North America, and Central and South America. In Japan, shellfish are essential ingredients, with an average household consuming approximately 2600 g of shellfish per year [19]. Scallops are one of the preferred shellfish species in Japan, with consumption reaching approximately 560 g per household [19]. Scallop adductor muscles are commonly used in sashimi, grilling, and simmering, and the scallop mantle, ovaries, and testes are often boiled and smoked in Japan. 

It is essential to thoroughly elucidate toxicity in mice to investigate whether novel toxins in scallops may affect human health. In the present study, to determine the characteristics of mantle tissue toxicity, the stability of the mantle toxin against digestive enzymes, the conditions under which mantle tissues exhibit toxicity, and toxicity in the small intestine were investigated.

## 2. Materials and Methods

### 2.1. Materials

Commercially available cultured 2-year-old scallops (*Patinopecten yessoensis*) harvested from Mutsu Bay (Aomori), Funka Bay (Hokkaido), Tokoro (Hokkaido), and Saruhutsu (Hokkaido), Japan, were purchased during different seasons (spring, summer, autumn, and winter). The mantle tissues used in the experiment were randomly selected from scallops. Unless otherwise stated in the figure legends, scallops were obtained from Mutsu Bay (Aomori, Japan). Smoked mantle products were purchased from various companies.

### 2.2. Extract from the Scallop Mantle Tissue

Mantle tissue was isolated, washed with deionized water, lyophilized, and ground in a mill. The product (20 g) was suspended in 500 mL of 20 mM Tris hydrochloride (Tris-HCl (Fujifilm Wako Co. Ltd., Osaka, Japan), pH 7.5) and centrifuged at 12,000× *g* for 15 min at 4 °C. The supernatant was used as the mantle extract [11,12,13]. 

### 2.3. Stability of Toxicity of the Mantle Tissue or Mantle Extract

To evaluate the stability of toxic substances in the mantle tissue, the tissue was treated under various conditions used in the manufacturing of processed foods. Mantle tissue was treated with 1 mM hydrochloric acid (HCl) or 1 mM sodium hydroxide (NaOH) at 25 °C for 6 h, rinsed with deionized water, and used for toxicity evaluation. Additionally, mantle tissue was treated with 1 mM hydrogen peroxide (H_2_O_2_) for 6 h or 1 mM dithiothreitol (DTT) for 6 h at 25 °C for sterilization and reduction. The mantle tissues were rinsed with deionized water and used for the toxicity evaluation. Furthermore, the mantle tissue was treated at 100 °C for 2 h, rinsed with deionized water, and used for toxicity evaluation.

To investigate the stability of the toxicity of the mantle extract, the pH was adjusted to 2 using 1 M HCl or to 11 using 1 M NaOH, and then maintained at 25 °C for 6 h. After adjusting the pH of each solution to 7, dialysis was performed using deionized water. The denatured proteins were removed through centrifugation at 12,000× *g* for 15 min at 4 °C, and the supernatant was freeze-dried and used for toxicity evaluation. The mantle extract was also treated at 100 °C for 2 h. After removing denatured proteins through centrifugation at 12,000× *g* for 15 min at 4 °C, the supernatant was freeze-dried and used for toxicity evaluation.

### 2.4. Stability of Toxicity of the Mantle Extract against Pepsin or Pancreatin Treatments

To evaluate the stability of toxicity of the mantle extract against digestive enzymes, the mantle extract was treated with pepsin (50:1 [*w*/*w*]) at pH 2 at 37 °C overnight. Additionally, the mantle extract was treated with pancreatin (50:1 [*w*/*w*]) at pH 7 at 37 °C overnight. After treatment, each sample was freeze-dried and subjected to a toxicity evaluation. Samples containing only proteases without mantle extract were used as controls.

### 2.5. Evaluation of Toxicity of Mantle Tissues

Four-week-old male Institute of Cancer Research (ICR) mice were purchased from CLEA (Tokyo, Japan). The mice were housed individually in a room maintained at 22 °C. Three or four mice were housed in together in cages and maintained at 22 °C with free access to water and food. The mice were acclimatized for at least 7 d and used for each experiment. Mice were fed either a normal AIN-93G-based diet (control diet) or an AIN-93G-based diet containing 1% mantle tissue or 1–20% mantle extract at 4 g/d for 4–12 weeks [11]. The protein, carbohydrate, and lipid concentrations in the mantle extract were determined using a bicinchoninic acid (BCA) protein assay kit (Thermo Fisher Scientific, Waltham, MA, USA), the phenol–sulfate method, and the method of Bligh and Dyer [20], respectively. The protein, carbohydrate, lipid, and fiber contents of the mantle tissue were 30%, 50%, 5%, and 1%, respectively. The diet compositions are shown in Table 1 and Table 2. The toxicity to scallops under various conditions was assessed by measuring the reduction in food intake and monitoring the mortality of mice. Each group consisted of three mice. Unless otherwise stated in the figure legends, either mice fed a control diet or those fed a 1% mantle diet were used as controls for statistical analysis in each experiment. The amount of mantle extract or tissue in the diet was determined based on the methods of previous studies [11] and preliminary experiments. 

The small intestine and cecum were isolated 3 weeks after the mice started consuming a diet containing mantle tissue or a control diet. Small intestine (1 mg) was homogenized in 20 mL of deionized water and centrifuged at 12,000× *g* for 15 min at 4 °C. The supernatant was used as the small intestine extract.

Animal experiments were conducted in accordance with the Guidelines for Experimental Animal Care issued by the Office of the Prime Minister of Japan and the Muroran Institute of Technology. All experiments were approved by the Committee on Ethics, Care, and Use of Animal Experiments of the Muroran Institute of Technology (Permit Number: H29-KS04). The health status of all the mice was monitored by assessing their daily food and water intake and observing their appearance during the experimental period. 

### 2.6. Electrophoresis 

Sodium dodecyl sulfate–polyacrylamide gel electrophoresis (SDS-PAGE) was performed as described previously [14]. Briefly, freeze-dried samples (1 mg) of pepsin- and pancreatin-treated mantle extract were suspended in 100 μL of SDS sample buffer containing 2% SDS, 20 mM Tris-HCl (pH 7.5), 1 mM 2-mercaptoethanol, 10% glycerol, and bromophenol blue. After heating the samples at 100 °C for 5 min, they were centrifuged at 12,000× g for 5 min, and the supernatant was subjected to SDS-PAGE. SDS-PAGE was performed using an NA-1010 mini slab gel electrophoresis unit (Nihon Eido, Tokyo, Japan) according to the method described by Leammli [21].

### 2.7. Real-Time Polymerase Chain Reaction

Total RNA was extracted from small intestinal tissues using the RNAiso Plus Kit (Takara, Shiga, Japan). First-strand cDNA was synthesized using 10 μg of RNA and an oligo (dT) primer. Real-time polymerase chain reaction (PCR) was performed using iTaq Universal SYBR Green Supermix (Bio-Rad, Hercules, CA, USA) [22], with 25 ng of cDNA template and gene-specific primers (200 nM) targeting various genes [22], including actin, Mn-superoxide dismutase (SOD), Cu, Zn-SOD, catalase, heme oxygenase (HO)-1, activating transcription factor 4 (ATF4), CCAAT-enhancer-binding protein homologous protein (CHOP), binding immunoglobulin protein (BiP), interleukin (IL)-1β, IL-6, tumor necrosis factor (TNF)-α, transforming growth factor (TGF)-β, cyclooxygenase (COX)-2, and inducible nitric oxide synthase (iNOS) (Table 3). Cycling conditions were set as follows: 40 cycles of 95 °C for 5 s, and 60 °C for 1 min. Target gene expression was normalized to the mean expression level of β-actin using the comparative Ct method.

### 2.8. Measurement of Lipid Peroxidation

Malondialdehyde levels, a marker of oxidative stress, were evaluated using thiobarbituric acid as previously described [22,23]. Briefly, a mixture containing intestine extract and 50% trichloroacetic acid (TCA) was prepared and centrifuged at 14,000× *g* for 1 min at 25 °C. Thiobarbituric acid (TBA) solution (0.67%) was incubated with the supernatant at 100 °C for 1 min, and absorbance at 540 nm was measured. 

### 2.9. Measurements of Short-Chain Fatty Acids (SCFAs)

The concentration of SCFAs in the cecal contents was determined using gas chromatography (GC) [24]. Briefly, 10 mg of cecal content was vigorously vortexed five times with deionized water (500 μL). After the mixture was centrifuged at 12,000× *g* for 1 min at 25 °C, 5 μL of 20 mM 2-ethylbutyric acid (internal standard) solution and 5 μL of 35% HCl solution were added to the supernatant. Diethyl ether (200 μL) was added to the mixed solution and centrifuged at 12,000× *g* for 1 min at 25 °C. The upper layer was used to measure the SCFA contents using GC with a GC-4000 GC system (Shimadzu, Kyoto, Japan) equipped with a flame ionization detector (FID) and a capillary column (length, 15 m; internal diameter, 0.32 mm; and film thickness, 0.25 mm). The temperature program was set as follows: 160 °C for 2 min, followed by a linear gradient from 160 °C to 300 °C at a rate of 20 °C/min. The total analysis time was 10  min. 

### 2.10. Statistical Analysis

The toxicity of scallops under various conditions was assessed based on the decrease in food intake. Data on food intake from three mice per group are expressed as the mean ± standard deviation (SD). Food intake of the dead mice was recorded as zero for the calculations.

Statistical analysis was performed using one-way analysis of variance (ANOVA), followed by the Tukey–Kramer multiple comparison test using BellCurve for Excel ver. 2.15 (SSRI, Tokyo, Japan). Statistical significance was set at *p* < 0.05.

Data from real-time PCR, SCFA contents, and malondialdehyde (MDA) contents in the small intestine were obtained using twelve mice and expressed as the mean ± SD. Statistical analysis was performed using Student’s two-tailed *t*-test. Statistical significance was set at *p* < 0.05. Each experiment was performed at least twice to ensure reproducibility.

## 3. Results

### 3.1. Stability of Toxicity in the Mantle Tissue or Extract

Toxicity in mice was evaluated by comparing the decrease in food intake, reflecting the toxicity of the mantle tissue, as described previously [11]. The stability of the mantle tissue toxicity was examined under various conditions. Tissue toxicity was assessed by treating the mantle tissue with 1 mM HCl or 1 mM NaOH for 6 h at 25 °C (Figure 1a). Treatment with 1 mM HCl or 1 mM NaOH resulted in a decrease in food intake starting in the third weeks and the mice died by the fourth week. There was no significant difference in food intake between weeks 1 and 4 between mice fed the diet containing mantle tissue treated with 1 mM HCl, 1 mM NaOH, and the mantle diet. Next, toxicity was investigated following heating, sterilization, and reduction treatments of the mantle tissue (Figure 1b,c). Mice fed a diet containing mantle treated with 1 mM H_2_O_2_ or 1 mM dithiothreitol exhibited a decrease in food intake starting in the third week and died in the fourth or fifth weeks, similar to mice fed the mantle diet. After treating the mantle tissue at 100 °C for 20 min, food intake did not change significantly between mice fed the diet containing heat-treated mantle and those fed the mantle diet at the first, second, fourth, and fifth weeks.

These results suggest that the toxic substances were stable within the mantle tissue.

Furthermore, we examined the stability of mantle extract toxicity (Figure 2a). 

In mice fed a diet containing NaOH-treated mantle extract, a decrease in food intake was observed in the fourth and fifth weeks, which recovered after the sixth week; the mice did not die until the seventh week. 

After heating, the toxicity of the mantle extract was reduced, whereas treatment with 1 mM HCl had no effect on mantle extract toxicity. Food intake did not change significantly during the first to fourth weeks between mice fed the diet containing HCl-treated mantle extract and those on the mantle diet. This result was consistent with our previous study [14], which reported that toxic substances remain stable under acidic conditions. We also examined the stability of the mantle extract against the digestive enzymes pepsin and pancreatin. Treatment with pepsin or pancreatin degraded the mantle extract; however, the protease-treated mantle extract exhibited a decrease in food intake in mice starting in the second or third weeks and died by the fourth week (Figure 2b,c). 

Finally, the toxicities of three smoked mantles produced by various companies were evaluated. The mantle diet showed a decrease in food intake starting in the second week, and mice died in the fourth week. One (Product A) of the processed mantle food exhibited a decrease in food intake and mice died after 8 weeks, although the toxicity was weaker than that of the mantle tissue. A significant difference in food intake from the fourth week was observed between mice fed a diet containing Product A and those fed the mantle diet, whereas products B and C showed no toxicity (Figure 2d).

### 3.2. Toxicity of the Mantle Tissue under Different Conditions

To examine whether toxic substances in the mantle tissue exhibited acute toxicity, mice were fed diets containing different amounts of mantle extract (1%, 3%, 5%, 10%, and 20%) (Figure 3a). Mice fed a diet containing 20% mantle extract showed a decrease in food intake starting at the second week and died by the third week. Mice fed diets containing 1% and 3% mantle extract exhibited decreased food intake after the second week and died by the fourth week. No significant difference in food intake during the first to third weeks was observed between mice fed a diet containing 1% and 20% mantle tissue. These results indicate that, even when consuming a large amount of mantle tissue, acute toxicity is not exhibited.

To investigate whether a toxic substance accumulated in tissues, we examined the toxicity when mantle tissue was administered to mice every other day, every 2 days, and once weekly (Figure 3b). When administered to mice daily, food intake began to decrease in the second week, and the mice died in the third week. The decrease in food intake of mice fed every other day was significantly delayed compared with that in mice fed daily, and they died in the fifth week. However, when mantle tissue was administered every 2 or 7 d, food intake did not decrease, and mouse mortality did not occur for up to 12 weeks. This suggests that mantle tissue must be continuously ingested to observe its toxic effects.

Next, we investigated whether toxicity was observed after the mice were fed a diet containing 1% mantle tissue for two weeks, followed by switching to a control diet (Figure 3c). Even after switching to the control diet, a decrease in food intake was observed, similar to that observed in mice fed the mantle diet. This indicates that organ impairment may occur when a reduction in food intake is observed.

Finally, we determined whether the toxicity of scallop mantle tissue varied depending on the location and season of collection (Figure 4a,b). All scallops collected from any location or season exhibited toxic effects on mice. No significant differences in the decrease in food intake were observed between the various collection locations and seasons. 

### 3.3. Toxicity of the Mantle Tissue in the Small Intestine

A previous study has demonstrated that a diet containing scallop mantle tissue leads to liver and kidney injury. In the present study, we focused on mantle tissue toxicity in the small intestine. Mice fed a diet containing 1% mantle tissue for 3 weeks showed a change in the color of the small intestine (Figure 5a). Levels of SCFAs associated with intestinal barrier function were examined. In mice fed a diet containing mantle tissue, there was a tendency towards decreased SCFA levels, with a significant decrease in propionic acid (Figure 5b). Subsequently, the small intestinal tissue was assessed for oxidative damage and inflammation.

In the small intestines of mice fed mantle tissue, there were significant increases in the levels of peroxidized lipids (Figure 6a), and the expression levels of the antioxidant enzymes catalase, Cu, Zn-SOD, and Mn-SOD increased (Figure 6b). Additionally, to determine whether endoplasmic reticulum (ER) stress occurred, the expression levels of CHOP, BiP, and ATF4, which are upregulated during ER stress, were investigated (Figure 6c). In mice fed the mantle diet, the expression level of BiP increased significantly, and those of ATF4 and CHOP tended to increase. The expression levels of iNOS, COX2, and TGF-β, which are inflammatory cytokines, also significantly increased (Figure 6d), suggesting that the ingested mantle tissue causes oxidative stress, ER stress, and inflammation. 

## 4. Discussion

In the present study, we compared the mantle tissue toxicity in mice by assessing food intake as an indicator of toxicity. A decrease in food consumption was frequently noticed within 2–3 weeks, resulting in death of the mice between the third and fifth weeks. However, the intensity of toxicity and food intake varied considerably between the experiments. Although the exact reasons remain unclear, this variability may be significantly influenced by the conditions of the purchased mice and the state of the scallops [25], even when subjected to identical conditions. Therefore, in this study, experiments were performed using scallops and mice purchased on the same day.

Our previous study showed that the toxic substance in mantle tissue is a protein complex [14]. Although the toxicity of the mantle extract was lost when exposed to heat and alkaline treatments, the toxicity of the mantle extract exhibited resistance to pepsin treatment under acidic conditions and to pancreatin treatment; the toxicity of the mantle tissue remained unaffected by various treatments. This result is supported by the fact that one of the smoked mantle products maintained its toxicity, although the toxicity was reduced. Although most proteinaceous toxins are not toxic when administered orally, due to their low absorption from the small intestine and degradation by digestive enzymes [26,27,28], mantle toxins are toxic even after oral administration. 

The ricin toxin exhibits toxicity even after oral administration [18]. Similar to mantle toxins, ricin is also reported to be highly stable in the presence of acids, heat, and digestive enzymes [29]. Additionally, the oral ingestion of ricin can lead to gastrointestinal injury [30]. Although mantle toxins do not show acute toxicity, several similar characteristics are observed with ricin. It would be interesting to further investigate the similarities between mantle toxins and ricin.

Several studies have reported that resistant proteins are resistant to digestive enzymes and behave as dietary fibers, similarly to cellulose, pectin, gum, and lignin [31,32]. Additionally, resistance proteins have been reported to influence the gut microbiota [18]. Murray et al. [33] reported that pigs fed a diet containing highly resistant proteins experienced weight loss and exhibited changes in their gut microbiota. Changes in the gut microbiota can lead to alterations in SCFAs [34], which play a crucial role in maintaining intestinal barrier function by preventing intestinal inflammation and oxidative stress [35,36]. In mice fed a diet containing mantle tissue, SCFA levels significantly decreased, in addition to inflammation, ER stress, and oxidative damage. Toxins in the mantle tissue may also influence the gut microbiota, leading to a decrease in SFCAs. Some toxic substances, such as the T2 toxin, can induce toxicity by causing ER stress and inflammation, leading to the destruction of the small intestinal mucosa [37]. Toxins produced by *Bacteroides fragilis* induce inflammation in the small intestine and disrupt barrier function [38]. Similarly, these toxic substances and toxins in the mantle tissue appear to disrupt the barrier function of the small intestine. 

In a previous study, a complex of gelsolin-like proteins and actin was isolated from scallops as a novel toxin that acts on the actin cytoskeleton in cells, causing changes in cell morphology and exhibiting toxicity [14]. *Clostridium* toxin A, which causes inflammation in the human small intestine, affects the actin cytoskeleton of small intestinal epithelial cells, resulting in barrier dysfunction [39,40]. Toxins in mantle tissue may also disrupt barrier function by affecting the actin cytoskeleton of small intestinal epithelial cells. A previous study showed that mice fed diets containing mantle tissue exhibited liver and kidney injuries. The ingestion of food containing mantle tissue may initially disrupt barrier function by displaying toxicity against the epithelial cells of the small intestine, potentially leading to the incorporation of proteinaceous toxins into the body. Further investigations are needed to explore the uptake of these toxic substances into the body.

When the mice were fed a diet containing 20% or 1% mantle extract, no distinct decrease in food intake was observed. This result indicates that even the ingestion of large amounts of mantle tissue does not cause acute toxicity. The amount of toxins absorbed into the small intestinal tissues may be limited. Furthermore, when mantle tissue was ingested every other day, toxicity was observed. However, when ingested every 2 d, no toxicity was observed. It may be necessary for a certain quantity of toxic substances to be present within the small intestine and continuously act on intestinal epithelial cells to cause toxicity. The toxicity of mantle tissues occurs with continuous ingestion; therefore, toxin in the mantle tissue does not appear to accumulate gradually in the intestines, liver, and kidneys. 

The toxicity of the scallop mantle tissue was not dependent on the location or season of collection. The expression levels of a previously identified toxic substance, gelsolin-like protein, may not vary significantly with location or season. In this study, we focused on scallop mantle tissue. Our previous study reported that the scallop ovarian tissue, which is often consumed in Japan, is toxic. This toxin may also be present in other shellfish species. *Lingula anatina* and *Mercenaria mercenaria* have gelsolin-like proteins with high homology values of 86% and 82%, respectively, to scallop gelsolin-like proteins, as shown in the BLAST search. These shellfishes are also consumed in Japan. Therefore, further investigations into various processed products and shellfish are required.

Whether the ingestion of mantle tissue causes toxicity in humans has not been reported, likely because feeding mantle tissue does not cause acute toxicity, and the continuous ingestion of mantle tissue is required to induce toxicity. However, the possibility that the chronic ingestion of mantle tissue affects human health, including intestinal inflammation, and kidney and liver injury, cannot be excluded. Currently, a search for various substances that can alleviate this toxicity and elucidate the pathways necessary for the expression of toxicity in mantle tissues is underway.

## 5. Conclusions

Toxicity of the mantle tissue persists even after acid, base, and heat treatments, indicating the marked stability of the toxic substance in the mantle tissue. Toxicity was observed when mantle tissue was consumed continuously, and the ingestion of a large amount of mantle tissue did not result in acute toxicity. Additionally, changing to a control diet after feeding the mice mantle tissue led to a decrease in food intake and death. The present results suggest that the continuous ingestion of mantle tissue triggers inflammation, ER stress, and oxidative stress in the small intestine, leading to the incorporation of toxic substances into the liver, kidneys, and other parts of the body. In the future, epidemiological research should be performed to determine whether the intake of scallop mantle and ovaries, which contain new potential toxins, influences human health, specifically its correlation with the incidence of intestinal inflammation, liver disease, and kidney disease.

## Figures and Tables

**Figure 1 foods-12-03224-f001:**
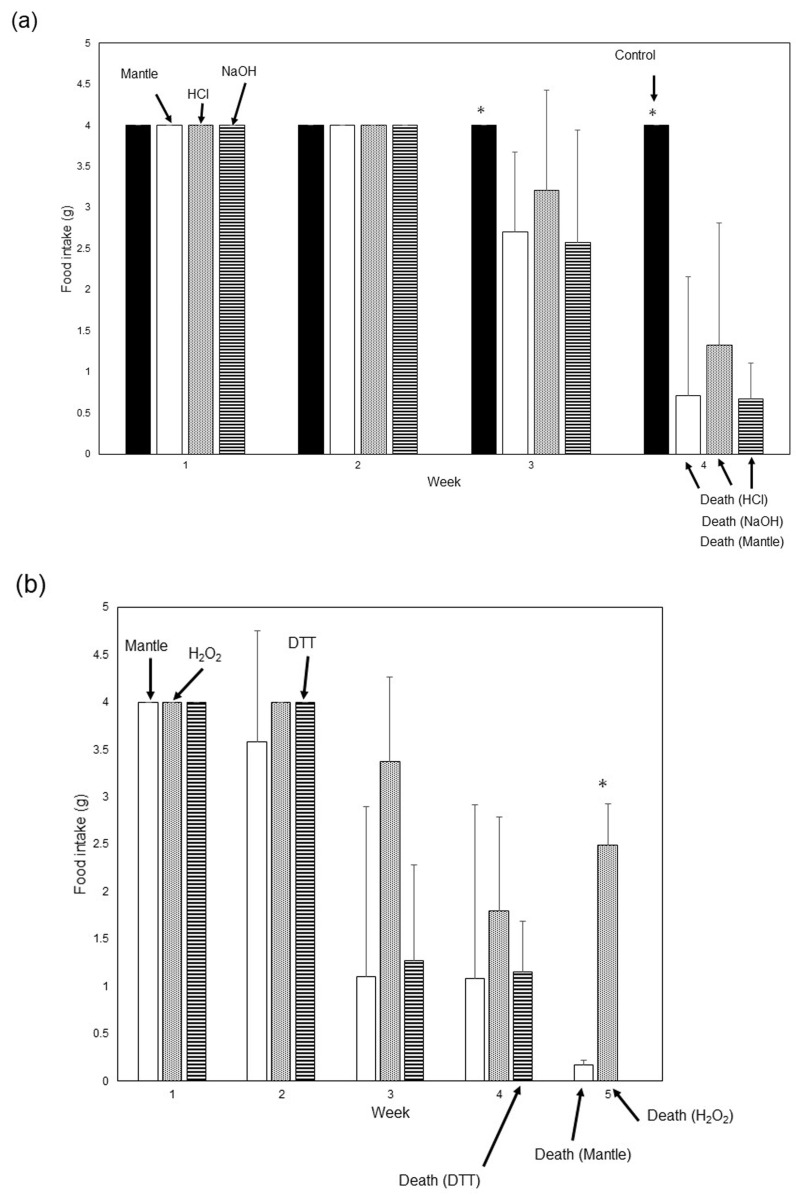
Toxicities of the NaOH-, HCl-, H_2_O_2_-, DTT-, and heat-treated mantle tissues. (**a**) Diets containing 1% HCl-treated mantle tissue (dotted bar), 1% NaOH-treated mantle tissue (striped bar), 1% non-treated mantle tissue (open bar), or control diet (closed bar) were fed to mice. (**b**) Diets containing 1% H_2_O_2_-treated mantle tissue (dotted bar), DTT-treated mantle tissue (striped bar), or 1% non-treated mantle tissue (open bar) were fed to mice. (**c**) Diets containing 1% heat-treated mantle tissue (dotted bar) or 1% non-treated mantle tissue (open bar) were fed to the mice. The food intake of mice during each week is shown. Data from three mice were combined to obtain means; the bars indicate the SD. * *p* < 0.05 relative to the mantle diet (using ANOVA).

**Figure 2 foods-12-03224-f002:**
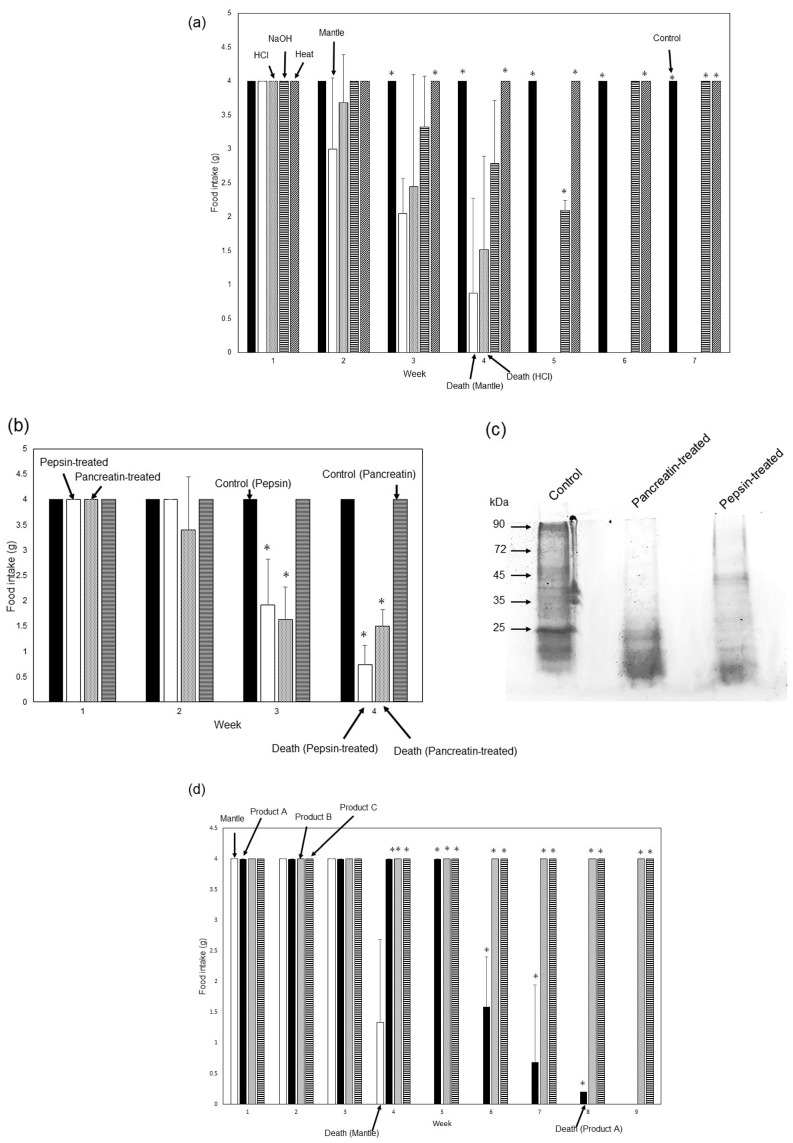
Toxicities of the NaOH-, HCl-, and heat-treated mantle extracts. (**a**) Diets containing 1% HCl-treated mantle extract (dotted bar), 1% NaOH-treated mantle tissue (striped bar), 1% non-treated mantle tissue (open bar), 1% heat-treated mantle extract (hatched bar), or control diet (closed bar) were fed to mice. Data were combined from three mice to calculate means; the bars show the SD. * *p* < 0.05 relative to the mantle diet (using ANOVA). (**b**) Diets containing 1% pepsin-treated mantle extract (open bar), 1% pancreatin-treated mantle extract (dotted bar), control diet containing pepsin (closed bar), or control diet containing pancreatin (striped bar) were fed to mice. Data were combined from three mice to calculate means; the bars show the SD. * *p* < 0.05 relative to the control diet (using ANOVA). (**c**) SDS-PAGE of mantle extract, pepsin-treated mantle extract, and pancreatin-treated mantle extract. (**d**) Diets containing 1% smoked mantle product A (closed bar), 1% smoked mantle product B (dotted bar), 1% smoked mantle product C (striped bar), or 1% mantle tissue (open bar) were fed to mice. The food intake of each week is shown. Data were combined from three mice; the bars show the SD. * *p* < 0.05 relative to the mantle diet (using ANOVA).

**Figure 3 foods-12-03224-f003:**
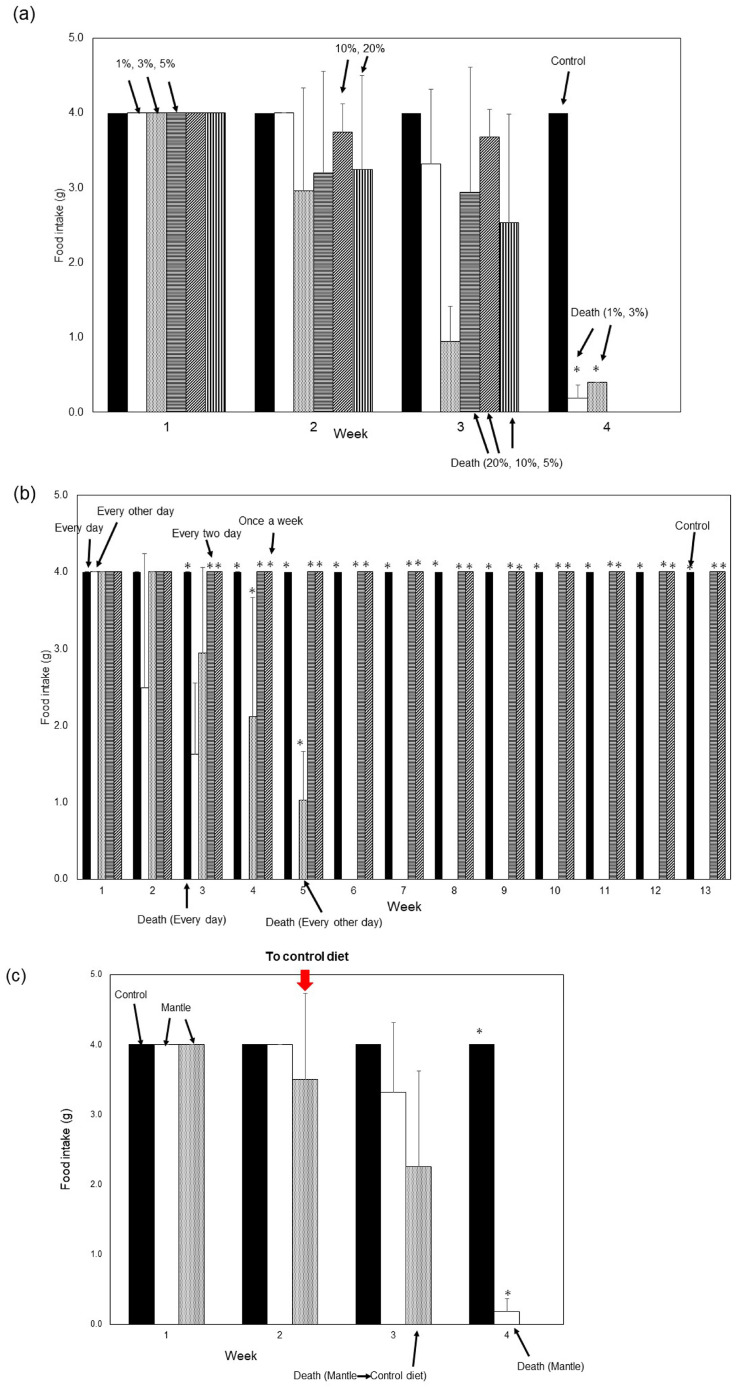
Toxicities of the mantle extract or tissue under various conditions. (**a**) Mice were fed diets containing 1% mantle extract (open bar), 3% mantle extract (dotted bar), 5% mantle extract (striped bar), 10% mantle extract (hatched bar), 20% mantle extract (vertical striped bar), or without mantle extract (closed bar). The food intake of each week is shown. Data from three mice were combined to calculate means; the bars show the SD. * *p* < 0.05 relative to 20% mantle diet (using ANOVA). (**b**) Diets containing 1% mantle tissue were fed to mice daily (open bar), every other day (dotted bar), every 2 days (striped bar), and every 7 days (hatched bar), and a control diet (closed bar) was fed to mice in the control group. The food intake of each week is shown. Data from three mice were combined to calculate means; the bars show the SD. * *p* < 0.05 relative to daily (using ANOVA). (**c**) The mantle diet (open bar) or the control diet (closed bar) were continued to be fed the mice (open bar). After diets containing 1% mantle tissue were fed to mice for two weeks, followed by switching to the control diet (dotted bar). Data from three mice were combined to calculate means; the bars show the SD. * *p* < 0.05 relative to the mantle diet (using ANOVA).

**Figure 4 foods-12-03224-f004:**
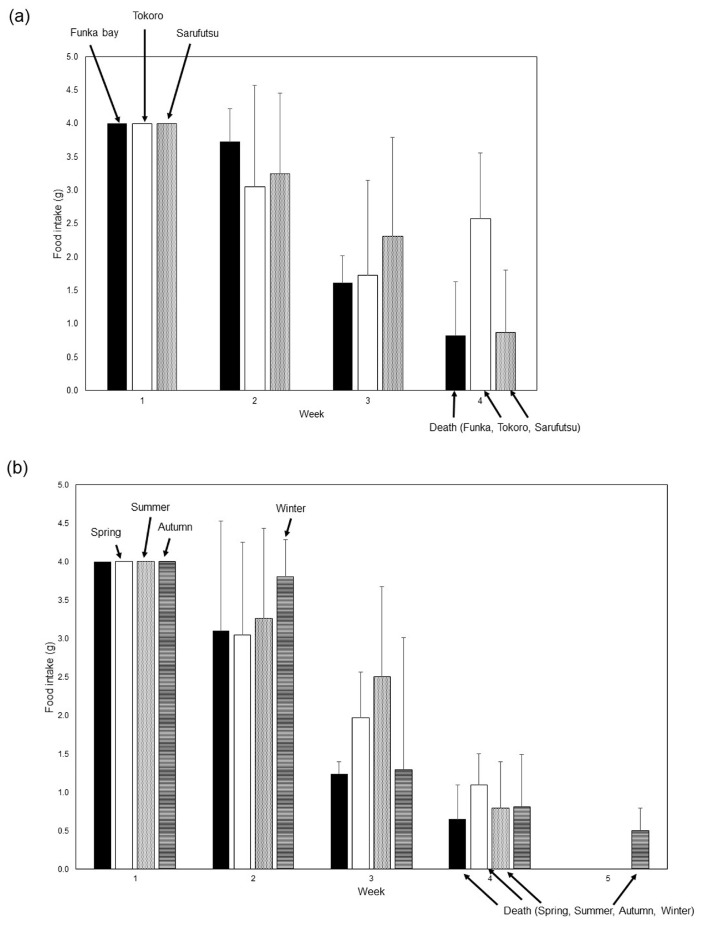
Toxicities of the scallop mantle tissue from various locations and seasons. (**a**) Mice were fed diets containing 1% mantle tissue from Funka Bay (Hokkaido) (closed bar), 1% mantle tissue from Tokoro (Hokkaido) (open bar), and 1% mantle tissue from Saruhutsu (Hokkaido) (dotted bar). (**b**) Mice were fed diets containing 1% mantle tissue from scallop harvested from Mutsu Bay (Aomori) in spring (closed bar), in summer (open bar), autumn (dotted bar), or winter (striped bar). Data from three mice were combined to calculate means; the bars show the SD.

**Figure 5 foods-12-03224-f005:**
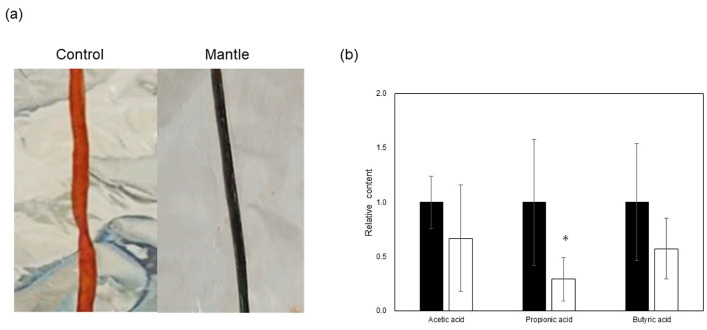
Toxicity of the mantle tissue in the small intestine. (**a**) Representative photograph of small intestines of mice fed control diet (left) or a diet containing 1% mantle tissue (right). (**b**) SCFA concentrations in cecal contents. Control diet (closed bar) or a diet containing 1% mantle tissue (open bar). The values of 12 mice are presented as the means ± SD. * *p* < 0.05 relative to control diet (Student’s *t*-test).

**Figure 6 foods-12-03224-f006:**
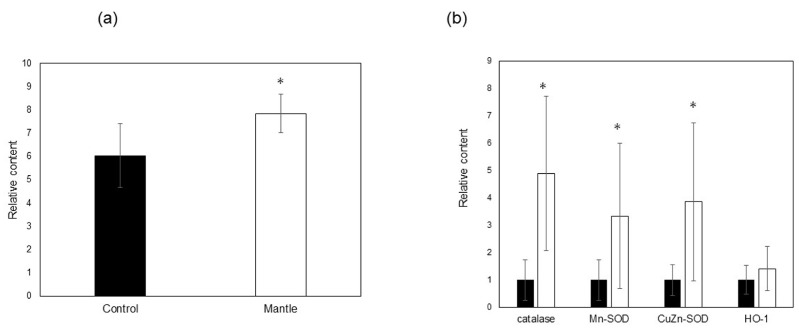
Changes in oxidative stress, inflammation, and ER stress in the small intestines of mice fed the control and mantle diets. (**a**) MDA content in the small intestine. Lipid peroxidation in the small intestines of mice fed either the control (closed bar) or mantle diets (open bar) is expressed as the MDA content. (**b**–**d**) Changes in the expression of enzymes in the small intestines of mice fed the control and mantle diets were measured using real-time PCR. (**b**) Antioxidant enzymes, (**c**) ER-stress-related proteins, and (**d**) inflammation-related proteins. The values of 12 mice are presented as the means ± SD. Asterisks indicate statistically significant differences relative to the control diet (** p* < 0.05).

**Table 1 foods-12-03224-t001:** Diet composition of control diet and mantle diet containing 1% mantle tissue or mantle extract, and treated mantle tissue or mantle extract.

	Control Diet (%)	1% Mantle Diet (%)
Casein	20.3	20
Corn starch	15.55	15
Cellulose	5.1	5
Mineral mixture	3.5	3.5
Vitamin mixture	1	1
L-cysteine	0.3	0.3
Choline bitartrate	0.2	0.2
Sucrose	50	50
Soybean oil	5.05	5
Mantle tissue or extract, treated mantle or extract	0	1
	101	101

**Table 2 foods-12-03224-t002:** Composition of the diet containing different amounts of mantle tissue and the corresponding control diets.

	Control Diet (%)	20% Mantle Diet (%)		Control Diet (%)	10% Mantle Diet (%)
Casein	26	20	Casein	23	20
Corn starch	26	15	Corn starch	20.5	15
Cellulose	7	5	Cellulose	6	5
Mineral mixture	3.5	3.5	Mineral mixture	3.5	3.5
Vitamin mixture	1	1	Vitamin mixture	1	1
L-cysteine	0.3	0.3	L-cysteine	0.3	0.3
Choline bitartrate	0.2	0.2	Choline bitartrate	0.2	0.2
Sucrose	50	50	Sucrose	50	50
Soybean oil	6	5	Soybean oil	5.5	5
Mantle tissue	0	20	Mantle tissue	0	10
	120	120		110	110
	**Control Diet (%)**	**5% Mantle Diet (%)**		**Control Diet (%)**	**3% Mantle Diet (%)**
Casein	21.5	20	Casein	20.9	20
Corn starch	17.75	15	Corn starch	16.65	15
Cellulose	5.5	5	Cellulose	5.3	5
Mineral mixture	3.5	3.5	Mineral mixture	3.5	3.5
Vitamin mixture	1	1	Vitamin mixture	1	1
L-cysteine	0.3	0.3	L-cysteine	0.3	0.3
Choline bitartrate	0.2	0.2	Choline bitartrate	0.2	0.2
Sucrose	50	50	Sucrose	50	50
Soybean oil	5.25	5	Soybean oil	5.15	5
Mantle tissue	0	5	Mantle tissue	0	3
	105	105		103	103

**Table 3 foods-12-03224-t003:** Primer sequences.

Gene Name	Sequence (5′ to 3′)
Actin	F-GGC TGT ATT CCC CTC CAT CG
	R-CCA GTT GGT AAC AAT GCC ATG T
Mn-SOD	F-GGC CAA GGG AGA TGT TAC AA
	R-GCT TGA TAG CCT CCA GCA AC
CuZn-SOD	F-CGG ATG AAG AGA GGC ATG TT
	R-CAC CTT TGC CCA AGT CAT CT
catalase	F-AGG TGT TGA ACG AGG AGG AG
	R-TGC GTG TAG GTG TGA ATT GC
HO-1	F-CAG GTG ATG CTG ACA GAG GA
	R-ACA GGA AGC TGA GAG TGA GG
ATF4	F-GAG CTT CCT GAA CAG CGA AGT
	R-TGG CCA CCT CCA GAT AGT CAT
CHOP	F-TCA CTA CTC TTG ACC CTG CG
	R-ACT GAC CAC TCT GTT TCC GT
Bip	F-CTG GGT ACA TTT GAT CTG ACT GG
	R-GCA TCC TGG TGG CTT TCC AGC CAT TC
IL-1α	F-GGG CCT CAA AGG AAA GAA TC
	R-TAC CAG TTG GGG AAC TCT GC
IL-6	F-AGA CTT CCA TCC AGT TGC CT
	R-CAG GTC TGT TGG GAG TGG TA
TGF-β	F-CAA CTT CTG TCT GGG ACC CT
	R-TAG TAG ACG ATG GGC AGT GG
TNF-α	F-ACG GCA TGG ATC TCA AAG AC
	R-GTG GGT GAG GAG CAC GTA GT
COX	F-AGA AGG AAA TGG CTG CAG AA
	R-GCT CGG CTT CCA GTA TTG AG
iNOS	F-CCC CGC TAC TAC TCC ATC AG
	R-CCA CTG ACA CTT CGC ACA AA

## Data Availability

The data used to support the findings of this study can be made available by the corresponding author upon request.

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
