# Peer review of "Characterization and Stability of a Novel Toxin in Scallop Mantle Tissue"

_foods, 2023, doi:10.3390/foods12173224_

Round 1

Reviewer 1 Report

This is an interesting study on toxins in scallop mantle tissue, however, for the article to be considered for publication it is necessary to make the following considerations:

1. Remove the word 'mantle tissue' from the keywords since the words that appear in the title should not be included.

2. The introduction is very deficient, since it should include a section that talks about the consumption of this type of crustacean worldwide and specifically in Japan. Also include the main toxins found and the percentage in which they are found.

3. It is necessary to highlight at the end of the introduction what the objectives of the work are.

4. Materials and methods. Describe the sampling criteria used to make this type of selection. Tables 1 and 2 should be improved in terms of resolution and adapt them to the standards of the journal.

5. In section 2.6. Electrophoresis should describe in more detail the technique and equipment used.

6. Results and discussion. The resolution of figures 1-6 should be improved and a more detailed description of it should be made. In addition, the discussion must be expanded since it is very concise and with few references.

An English review is required

Author Response

We wish to express our strong appreciation to the reviewers for their insightful comments on our paper. We feel the comments have helped us significantly improve the paper. We attach here our revised manuscript and point-by-point response to the reviewer’s comments.

This is an interesting study on toxins in scallop mantle tissue, however, for the article to be considered for publication it is necessary to make the following considerations:

(A) Remove the word 'mantle tissue' from the keywords since the words that appear in the title should not be included.

(Response A)In accordance with reviewer’s comment, we removed the word 'mantle tissue' from the keywords.

(B). The introduction is very deficient, since it should include a section that talks about the consumption of this type of crustacean worldwide and specifically in Japan. Also include the main toxins found and the percentage in which they are found.

(Response B)In accordance with reviewer’s comment, we added the following text to “Introduction”

“Various types of shellfish, such as scallops, oysters, clams, and mussels, are widely consumed across Asia, Europe, North America, and Central and South America. In Japan, shellfish are also essential ingredients, with an average household consuming approximately 2,600 grams of shellfish per year. Among them, scallops are the preferred foods among the Japanese, with consumption reaching around 560 grams per household. Scallop adductor muscles are commonly used in dishes such as sashimi, grilling, and simmering. Scallop mantle ovaries, and testes, are also often eaten boiled and smoked in Japan. “

(C). It is necessary to highlight at the end of the introduction what the objectives of the work are.

(Response C)In accordance with reviewer’s comment, we added the following text to “Introduction”

“It is essential to thoroughly elucidate the toxicity to mice as a fundamental data for investigating whether novel toxins in scallops affect human health in future.”

(D). Materials and methods. Describe the sampling criteria used to make this type of selection. Tables 1 and 2 should be improved in terms of resolution and adapt them to the standards of the journal.

(Response D)In accordance with reviewer’s comment, we added the following text to “Materials”

“Commercially available cultured 2-year-old scallops (Patinopecten yessoensis) harvested from Mutsu Bay (Aomori), Funka Bay (Hokkaido), Tokoro (Hokkaido), and Saruhutsu (Hokkaido), Japan, were purchased during different seasons (spring, summer, autumn, and winter). The mantle tissue used in the experiment was randomly selected from the purchased scallops.”

(E). In section 2.6. Electrophoresis should describe in more detail the technique and equipment used.

(Response E)In accordance with reviewer’s comment, we added the following text to “Materials and methods, 2.6”

“Briefly, freeze-dried samples (1 mg) of mantle extract, pepsin- and pancreatin-treated mantle extract was suspended in 100 μL of SDS sample buffer containing 2% SDS, 20 mM Tris-HCl (pH 7.5), 1 mM 2-mercaptoethanol, 10% glycerol, and bromophenol blue. After heating the samples at 100 °C for 5 min, they were centrifuged at 12,000 g for 5 minutes, and the supernatant was subjected to SDS-PAGE. SDS-PAGE was performed according to the method of Leammli [15] using an NA-1010 mini slab gel
electrophoresis unit (Nihon Eido, Tokyo, Japan).”

(F). Results and discussion. The resolution of figures 1-6 should be improved and a more detailed description of it should be made. In addition, the discussion must be expanded since it is very concise and with few references.

(Response F)In accordance with reviewer’s comment, we improved the resolution of figure 1-6 and added a detail description in “Results” and “Discussion”.

Thank you again for your comments on our paper. We trust that the revised manuscript is suitable for publication. I want to make your useful advice for future research.

Reviewer 2 Report

Overall, the manuscript provides valuable insights into the stability and toxicity of a novel toxin in scallop mantle tissue and its effects on the small intestine. One of the major shortcomings of this manuscript is the absence of displaying the performed statistical processing (as specified in the Materials and Methods section), which could hinder transparency and proper interpretation of the results.

Specific comments:

Title: Characterization and Stability of a Novel Toxin in Scallop Mantle Tissue

Title: The title "Characterization and stability of a novel toxin in scallop mantle tissue" provides a clear indication of the study's focus. It accurately represents the main objectives and subject matter.

Abstract: The abstract provides a concise summary of the study's key findings. It briefly introduces the previous work, states the study's purpose, and outlines the main results. It effectively highlights the novelty of the toxin's subacute toxicity and its stability in mantle tissues. Overall, the abstract is well-written and informative.

Correction: However, the abstract could be improved by including more specific quantitative results.

Introduction: The introduction effectively provides background information on bivalve mollusks and their accumulation of toxins. It references well-known toxins and their acute toxicity, establishing the context for the novel toxin discovered in scallop mantle tissues. The rationale for the study is clearly presented, and the gaps in knowledge regarding proteinaceous toxins that exhibit toxicity after oral administration are well-defined. The introduction successfully sets the stage for the research.

Correction: Consider providing a clearer transition between the background information and the specific objectives of the study.

Materials and Methods: The material and methods section thoroughly describes the experimental procedures, including the materials used, sample preparation, stability evaluations, animal experiments, and data analysis methods. The section is well-structured and easy to follow.

Corrections: In the Methods section, some parts lack sufficient explanations, warranting more detailed information about the experimental design and controls. For instance, specific methods employed for real-time PCR and gas chromatography should be elaborated upon to enhance clarity and understanding. Additionally, providing additional information about the controls and replicates used in each experiment would strengthen the robustness of the study. It is essential to include more comprehensive descriptions of the specific procedures employed for each method, as the current information appears somewhat brief. Moreover, it is crucial to mention whether appropriate controls were utilized in each experiment to ensure the validity and reliability of the results.

The information provided in the manuscript regarding statistical analysis is insufficient for replication. It lacks crucial details such as:

·         Sample Size: The total number of samples or observations used in each group or condition should be specified.

·         Alpha Level: The significance level (alpha) used for determining statistical significance should be specified (e.g., p < 0.05).

·         One-tailed or Two-tailed Test: It should be mentioned whether the t-test was one-tailed or two-tailed.

Including these details will ensure that other researchers can accurately reproduce the statistical analysis and interpret the results appropriately.

Results: The results section presents the findings of the study. It discusses the stability of toxicity in the mantle tissue and extract under various conditions, including treatment with HCl, NaOH, H2O2, dithiothreitol, heating, and exposure to digestive enzymes. It indicates that toxic substances in the mantle tissue are stable and retain toxicity even after these treatments. The results also demonstrate that mantle tissue does not exhibit acute toxicity, as diets containing 1% and 20% mantle extract showed similar levels of toxicity. Additionally, the study reveals that ingested mantle tissue causes oxidative stress, ER stress, and inflammation in the small intestine.

Correction: Add more specific information about the quantitative results obtained, including statistical analyses and data variability.

One of the major shortcomings of this manuscript is the absence of displaying the performed statistical processing (as specified in the Materials and Methods section), which could hinder transparency and proper interpretation of the results.

Discussion: The discussion provides a comprehensive analysis of the study's results and effectively relates them to existing knowledge in the field. The potential influence of toxic substances on the gut microbiota, the role of the toxin in disrupting the barrier function of the small intestine, and the connection to actin cytoskeleton disruption are well-discussed. The authors also highlight the need for further investigations into other shellfish species and processed products. The discussion is coherent and informative.

Correction: However, the discussion would benefit from more specific references to the quantitative results presented in the results section to strengthen the arguments made.

Conclusions: The conclusion section should summarize the key findings of the study. It should emphasize the stability of mantle toxins in different conditions and the absence of acute toxicity in the mantle tissue. Furthermore, it should discuss the implications of the study's results on the consumption of scallop mantle tissue and potential health risks.

Correction: The current conclusions provide a general overview of the findings, but they could be strengthened by adding more specific and actionable insights.

Author Response

â‘¡We wish to express our strong appreciation to the reviewers for their insightful comments on our paper. We feel the comments have helped us significantly improve the paper. We attach here our revised manuscript and point-by-point response to the reviewer’s comments.

(A) Abstract: 

Correction: However, the abstract could be improved by including more specific quantitative results.

(Response A)In accordance with reviewer’s comment, we corrected a part of the “Abstract”.

(B) Introduction: 

Correction: Consider providing a clearer transition between the background information and the specific objectives of the study.

(Response B)In accordance with reviewer’s comment, we added the text including the objective of this study in “Introduction”.

“It is essential to thoroughly elucidate toxicity in mice to investigate whether novel toxins in scallops may affect human health. In the present study, to determine the characteristics of mantle tissue toxicity, the stability of the mantle toxin against digestive enzymes, the conditions under which mantle tissues exhibit toxicity, and toxicity in the small intestine were investigated.”

(C-1) Materials and Methods:

Corrections: In the Methods section, some parts lack sufficient explanations, warranting more detailed information about the experimental design and controls. For instance, specific methods employed for real-time PCR and gas chromatography should be elaborated upon to enhance clarity and understanding.

(Response C-1)In accordance with reviewer’s comment, we added the following explanations for real-time PCR and gas chromatography in “Materials and methods”.

“Total RNA was extracted from small intestinal tissues using the RNAiso Plus Kit (Takara, Shiga, Japan). First-strand cDNA was synthesized using 10 μg of RNA and an oligo (dT) primer. Real-time polymerase chain reaction (PCR) was performed using iTaq Universal SYBR Green Supermix (Bio-Rad, Hercules, CA, USA) [17], with 25 ng of cDNA template and gene-specific primers (200 nM) targeting various genes [17], including actin, Mn-superoxide dismutase (SOD), Cu, Zn-SOD, catalase, heme oxygenase (HO)-1, activating transcription factor 4 (ATF4), CCAAT-enhancer-binding protein homologous protein (CHOP), binding immunoglobulin protein (BiP), interleukin (IL)-1β, IL-6, tumor necrosis factor (TNF)-α, transforming growth factor (TGF)-β, cyclooxygenase (COX)-2, and inducible nitric oxide synthase (iNOS) (Table 2). Cycling conditions were set as follows: 40 cycles of 95 °C for 5 s, and 60 °C for 1 min. Target gene expression was normalized to the mean expression level of β-actin using the comparative Ct method.”

“The upper layer was used to measure the SCFA contents using GC with a GC-4000 GC system (Shimadzu, Kyoto, Japan) equipped with a flame ionization detector (FID) and a capillary column (length, 15 m; internal diameter, 0.32 mm; and film thickness, 0.25 mm). The temperature program was set as follows: 160 °C for 2 min, followed by a linear gradient from 160 °C to 300 °C at a rate of 20 °C/min. The total analysis time was 10  min.”

 (C-2) Additionally, providing additional information about the controls and replicates used in each experiment would strengthen the robustness of the study. It is essential to include more comprehensive descriptions of the specific procedures employed for each method, as the current information appears somewhat brief. Moreover, it is crucial to mention whether appropriate controls were utilized in each experiment to ensure the validity and reliability of the results.

(Response C-2)In accordance with reviewer’s comment, we added the following description about the controls used in each experiment in “Materials and methods 2.5”

“The toxicity to scallops under various conditions was assessed by measuring the reduction in food intake and monitoring mortality of mice. Each group consisted of three mice. Unless otherwise stated in the figure legends, either mice fed a control diet or those fed a 1% mantle diet were used as controls for statistical analysis in each experiment.”

(C-3) The information provided in the manuscript regarding statistical analysis is insufficient for replication. It lacks crucial details such as:

  • Sample Size: The total number of samples or observations used in each group or condition should be specified.
  • Alpha Level: The significance level (alpha) used for determining statistical significance should be specified (e.g., p < 0.05).
  • One-tailed or Two-tailed Test: It should be mentioned whether the t-test was one-tailed or two-tailed.

Including these details will ensure that other researchers can accurately reproduce the statistical analysis and interpret the results appropriately.

(Response C-3)In acoordance with reviewer’s comment, we added the detailed following explanations of statistical analysis to “Materials and methods, 2.10”

“2.10. Statistical analysis

The toxicity of scallops under various conditions was assessed based on the decrease in food intake. Data on food intake from three mice per group are expressed as mean ± standard deviation (SD). Food intake of the dead mice was recorded as zero for the calculations.

Statistical analysis was performed using one-way analysis of variance (ANOVA), followed by the Tukey–Kramer multiple comparison test using Excel Statistics software (SSRI, Tokyo, Japan). Statistical significance was set at p < 0.05.

Data from real time PCR, SCFA contents, and malondialdehyde (MDA) contents in the small intestine were obtained using twelve mice and expressed as the mean ± SD. Statistical analysis was performed using Student's two-tailed t-test. Statistical significance was set at p < 0.05.Each experiment was performed at least twice to ensure reproducibility.”

(D) Results:

Correction: Add more specific information about the quantitative results obtained, including statistical analyses and data variability.

(Response D)In experiments evaluating the toxicity of mantle tissue, a significance test was performed against food intake to quantify each experiment. In accordance with reviewer’s comment, we added the explanations of quantitative results in “Results”.

(E) Discussion:

Correction: However, the discussion would benefit from more specific references to the quantitative results presented in the results section to strengthen the arguments made.

(Response E)In accordance with reviewer’s comment, we added the text and references in “Discussion”.

(F) Conclusions: The conclusion section should summarize the key findings of the study. It should emphasize the stability of mantle toxins in different conditions and the absence of acute toxicity in the mantle tissue. Furthermore, it should discuss the implications of the study's results on the consumption of scallop mantle tissue and potential health risks.

Correction: The current conclusions provide a general overview of the findings, but they could be strengthened by adding more specific and actionable insights.

(Response F)In accordance with reviewer’s comment, we corrected “Conclusions” as follows.

“The toxicity of the mantle tissue persisted even after acid, base, and heat treatments, indicating the marked stability of the toxic substance in the mantle tissue. Toxicity was observed when mantle tissue was consumed continuously, and the ingestion of a large amount of mantle tissue did not result in acute toxicity. Additionally, changing to a control diet after feeding the mice mantle tissue led to a decrease in food intake and death. The present results suggest that the continuous ingestion of mantle tissue triggers inflammation, ER stress, and oxidative stress in the small intestine, leading to the incorporation of toxic substances into the liver, kidneys, and other parts of the body. In the future, epidemiological research should be performed to determine whether the intake of scallop mantle and ovaries, which contain new potential toxins, influences human health, specifically its correlation with the incidence of intestinal inflammation, liver disease, and kidney disease.”

Thank you again for your comments on our paper. We trust that the revised manuscript is suitable for publication. I want to make your useful advice for future research.

Reviewer 3 Report

The paper investigates an important issue for consumers of scallops.

The following revisions should be considered:

- I would not start an abstract with "we". This is stylistically extremely inelegant. Generally throughout of the text, I would also avoid the word "we", which is used excessively. It is self-evident that the research is done by the authors.

- keywords: use semicolon as separator

- Discussion: can you comment of the effects may be reversible when consumption is stopped at a certain time point? This could be relevant for humans, who probably only occasionally consume scallops.

- Conclusions: could you suggest what would be the next steps to study the relevance of the results for humans consuming scallops

See above, could be edited for style and clarity

Author Response

â‘¢We wish to express our strong appreciation to the reviewers for their insightful comments on our paper. We feel the comments have helped us significantly improve the paper. We attach here our revised manuscript and point-by-point response to the reviewer’s comments.

The paper investigates an important issue for consumers of scallops.

The following revisions should be considered:

(A) I would not start an abstract with "we". This is stylistically extremely inelegant. Generally throughout of the text, I would also avoid the word "we", which is used excessively. It is self-evident that the research is done by the authors.

(Response A)In accordance with reviewer’s comment, we have corrected the sentences in the text that used the word 'we'.

(B)  keywords: use semicolon as separator

(Response B)In accordance with reviewer’s comment, we corrected keywords.

Keywords: novel toxin; scallop; small intestine; subacute toxicity”

(C)- Discussion: can you comment of the effects may be reversible when consumption is stopped at a certain time point? This could be relevant for humans, who probably only occasionally consume scallops.

(Response C)We have conducted the experiments suggested by the reviewer. We have added the results in the Figure 3c. Even after changing to a control diet at a time point food intake started to reduce, the mouse died.

(D)- Conclusions: could you suggest what would be the next steps to study the relevance of the results for humans consuming scallops

(Response D)In accordance with reviewer’s comment, we added the following text in “Conclusions”.

“In the future, epidemiological research should be performed to determine whether the intake of scallop mantle and ovaries, which contain new potential toxins, influences human health, specifically its correlation with the incidence of intestinal inflammation, liver disease, and kidney disease.”

Thank you again for your comments on our paper. We trust that the revised manuscript is suitable for publication. I want to make your useful advice for future research.

Reviewer 4 Report

This manuscript deals with toxin in scallop mantle tissue. The manuscript is interesting but there are some minor issues:

Table 1 should be better explained – is this diet for mice? Please explain.

You showed the toxicity of the mantle toxin is the any treatment to reduce toxicity?

Author Response

â‘£We wish to express our strong appreciation to the reviewers for their insightful comments on our paper. We feel the comments have helped us significantly improve the paper. We attach here our revised manuscript and point-by-point response to the reviewer’s comments.

This manuscript deals with toxin in scallop mantle tissue. The manuscript is interesting but there are some minor issues:

(A) Table 1 should be better explained – is this diet for mice? Please explain.

(Response A)In accordance with reviewer’s comment, we added the explanation of Tables.

We divided into two tables (Table 1 and 2) and explained them separately for better understanding.

“Table 1. Diet composition of control diet and mantle diets containing 1% mantle tissue or mantle extract, and treated-mantle tissue or mantle extract.”

“Table 2. Composition of the diets containing different amounts of mantle tissue and the corresponding control diets.”

(B) You showed the toxicity of the mantle toxin is the any treatment to reduce toxicity?

(Response B)We added the following text in “Discussion”.

“Currently, a search for various substances that can alleviate this toxicity and elucidate the pathways necessary for the expression of toxicity in mantle tissues is underway.”

Thank you again for your comments on our paper. We trust that the revised manuscript is suitable for publication. I want to make your useful advice for future research.

Reviewer 5 Report

The current manuscript submitted to Foods journal reports on the potential toxicity of ingestion of mantle tissue to humans after the conscutive consumption. The hypothesis driven in the study might be of interest for the food scientists including those involved in nutrition. 

The manuscript needs a re-formulation. The authors refer consecutively to first person and the data/information involved must be re-organized. The concept of the study is in general well designed. However, the authors must pay attention to the attached comments. The manuscript lacks of Tables. Some data must be given in Tables.

In my opinion, the manuscript could also be sent to Nutrients MDPI. Finally, a graphical abastract would strengthen more the quality of this work.

Based on my overall comments, I suggest a major revision.

The English language must be improved. The authors must discuss better their findings and concept of the study. They must avoid the consecutive use of first person.

Author Response

⑤We wish to express our strong appreciation to the reviewers for their insightful comments on our paper. We feel the comments have helped us significantly improve the paper. We attach here our revised manuscript and point-by-point response to the reviewer’s comments.

(A) The current manuscript submitted to Foods journal reports on the potential toxicity of ingestion of mantle tissue to humans after the conscutive consumption. The hypothesis driven in the study might be of interest for the food scientists including those involved in nutrition. 

(Response A)The main purpose of this research paper is to investigate the effects of a novel shellfish toxin from scallops on mice. As for its effects on humans, further investigation will be performed using epidemiological studies and other methods in the future.

(B) The manuscript needs a re-formulation. The authors refer consecutively to first person and the data/information involved must be re-organized. The concept of the study is in general well designed. However, the authors must pay attention to the attached comments. The manuscript lacks of Tables. Some data must be given in Tables.

(Response B)In accordance with reviewer’s comments, I have corrected about the points indicated in your attached file. However, I couldn't find any comments regarding the Tables.

(C)In my opinion, the manuscript could also be sent to Nutrients MDPI. Finally, a graphical abastract would strengthen more the quality of this work.

(Response C)In accordance with reviewer’s comments, I added a graphical abstract.

Thank you again for your comments on our paper. We trust that the revised manuscript is suitable for publication. I want to make your useful advice for future research.

Round 2

Reviewer 1 Report

Minor editing of English language required

Minor editing of English language required

Author Response

Minor editing of English language required.

Response

We wish to express our strong appreciation to the reviewers for their insightful comments on our paper. We feel the comments have helped us significantly improve the paper. We attach here our revised manuscript and point-by-point response to the reviewer’s comments.

We have conducted English proofreading on this paper as per the attached file. We would appreciate it if you could review it.

Thank you again for your comments on our paper. We trust that the revised manuscript is suitable for publication. I want to make your useful advice for future researc

Reviewer 2 Report

From the author responses provided, it is evident that the authors have diligently worked towards addressing the majority of the reviewer's comments and subsequently refining their manuscript. However, it is notable that the authors have not explicitly tackled the reviewer's feedback concerning the need for a more seamless transition between the background information and the specific study objectives within the Introduction section. The presented text appears to remain consistent with the original version of the manuscript, without evident modifications aimed at enhancing this transition.

In the section discussing the statistical data analysis, the authors mention the following: "Statistical analysis was performed using one-way analysis of variance (ANOVA), followed by the Tukey–Kramer multiple comparison test using Excel Statistics software (SSRI, Tokyo, Japan)." The authors are kindly requested to clarify the specific software program they are referring to.

Author Response

We wish to express our strong appreciation to the reviewers for their insightful comments on our paper. We feel the comments have helped us significantly improve the paper. We attach here our revised manuscript and point-by-point response to the reviewer’s comments.

1) From the author responses provided, it is evident that the authors have diligently worked towards addressing the majority of the reviewer's comments and subsequently refining their manuscript. However, it is notable that the authors have not explicitly tackled the reviewer's feedback concerning the need for a more seamless transition between the background information and the specific study objectives within the Introduction section. The presented text appears to remain consistent with the original version of the manuscript, without evident modifications aimed at enhancing this transition.

(Response 1)

In accordance with reviewer’s comment, we added the following text and references to “Introduction” (Line 39-49)

“Scallop is one of the significant marine products in Hokkaido, Japan, resulting in the generation of approximately 300, 000 tons per year of scallop shells as industrial waste. For effective utilization of the scallop shells, bioactivities of organic components in scallop shells have been extensively investigated [6-10]. Considering that these organic components are secreted from the mantle tissue, the bioactivities of the mantle tissue have also been investigated. In the process of this research, subacute toxicity was observed in mice that fed a diet containing mantle tissue available on the market. Our previous study revealed that feeding mice a diet containing the mantle tissue of scallops led to increased levels of the liver damage markers aspartate aminotransferase and alanine aminotransferase, an increase in the kidney damage marker urea nitrogen, and subsequent death of the mice within a few weeks [11–13].”

(2) In the section discussing the statistical data analysis, the authors mention the following: "Statistical analysis was performed using one-way analysis of variance (ANOVA), followed by the Tukey–Kramer multiple comparison test using Excel Statistics software (SSRI, Tokyo, Japan)." The authors are kindly requested to clarify the specific software program they are referring to.

(Response 2)

In accordance with reviewer’s comment, we have added software program in the text.(line 245-246).

Thank you again for your comments on our paper. We trust that the revised manuscript is suitable for publication. I want to make your useful advice for future research.

Reviewer 5 Report

The revised version of the manuscript has been improved and can be accepted for publication.

Author Response

Thank you again for your comments on our paper.